Gene signature for prognosis in comparison of pancreatic cancer patients with diabetes and non-diabetes

Yang Mingjun yangmj@lut.cn 1
Song Boni lut8866@163.com 1 2
Liu Juxiang 3
Bing Zhitong 2 4
Wang Yonggang 1
Yu Linmiao 1
1 School of Life Science and Engineering, Lanzhou University of Technology , Lanzhou , Gansu , China
2 Institute of Modern Physics of Chinese Academy of Sciences , Lanzhou , China
3 Gansu Key Laboratory of Endocrine and metabolism, Department of Endocrinology, Gansu Provincial People’s Hospital , Lanzhou , Gansu , China
4 Evidence Based Medicine Center, School of Basic Medical Science of Lanzhou University, , Lanzhou , China
Suner Aslı
Electronic publication date: 2020 Nov 11
Publication date: 2020
Volume: 8
Electronic Location ID: e10297
Received 2019 Jun 17; Accepted 2020 Oct 13
Copyright: ©2020 Yang et al.
Copyright year: 2020
Copyright holder: Yang et al.
License: This is an open access article distributed under the terms of the Creative Commons Attribution License, which permits unrestricted use, distribution, reproduction and adaptation in any medium and for any purpose provided that it is properly attributed. For attribution, the original author(s), title, publication source (PeerJ) and either DOI or URL of the article must be cited.
License URL: https://creativecommons.org/licenses/by/4.0/

Keywords: PC, Diabetes, LASSO Cox regression, Prognosis index, Diabetes, LASSO Cox regression, Prognosis index

Funding: National Natural Science Foundation of China 81660581 Natural Science Foundation of Gansu Province, China 1606RJZA016 This project was supported by the National Natural Science Foundation of China (Grant No. 81660581) and by the Natural Science Foundation of Gansu Province, China (Grant No. 1606RJZA016). The funders had no role in study design, data collection and analysis, decision to publish, or preparation of the manuscript.

==============================
Background

Pancreatic cancer (PC) has much weaker prognosis, which can be divided into diabetes and non-diabetes. PC patients with diabetes mellitus will have more opportunities for physical examination due to diabetes, while pancreatic cancer patients without diabetes tend to have higher risk. Identification of prognostic markers for diabetic and non-diabetic pancreatic cancer can improve the prognosis of patients with both types of pancreatic cancer.

Methods

Both types of PC patients perform differently at the clinical and molecular levels. The Cancer Genome Atlas (TCGA) is employed in this study. The gene expression of the PC with diabetes and non-diabetes is used for predicting their prognosis by LASSO (Least Absolute Shrinkage and Selection Operator) Cox regression. Furthermore, the results are validated by exchanging gene biomarker with each other and verified by the independent Gene Expression Omnibus (GEO) and the International Cancer Genome Consortium (ICGC). The prognostic index (PI) is generated by a combination of genetic biomarkers that are used to rank the patient’s risk ratio. Survival analysis is applied to test significant difference between high-risk group and low-risk group.

Results

An integrated gene prognostic biomarker consisted by 14 low-risk genes and six high-risk genes in PC with non-diabetes. Meanwhile, and another integrated gene prognostic biomarker consisted by five low-risk genes and three high-risk genes in PC with diabetes. Therefore, the prognostic value of gene biomarker in PC with non-diabetes and diabetes are all greater than clinical traits (HR = 1.102, P-value < 0.0001; HR = 1.212, P-value < 0.0001). Gene signature in PC with non-diabetes was validated in two independent datasets.

Conclusions

The conclusion of this study indicated that the prognostic value of genetic biomarkers in PCs with non-diabetes and diabetes. The gene signature was validated in two independent databases. Therefore, this study is expected to provide a novel gene biomarker for predicting prognosis of PC with non-diabetes and diabetes and improving clinical decision.

Introduction

PC is an aggressive cancer of the digestive system, which is becoming a serious health problem worldwide. Overall survival for patients with pancreatic cancer is poor, mainly due to a lack of biomarkers to enable early diagnosis and a lack of prognostic markers that can inform decision-making, facilitating personalized treatment and an optimal clinical outcome (Siegel, Miller & Jemal, 2016). In most cases, type-II diabetes frequently occurs in patients with PC .Thus, it is considered to be an important risk factor for malignancy of PC (Huxley et al., 2005). However, non-diabetes PC patients have no early diagnosis indicator, which makes it more difficult to diagnose. In addition, PC with diabetes and without diabetes are very different in histopathology (Girelli et al., 1995) and molecular levels. Currently, many studies do not consider the difference between PC with diabetes and non-diabetes. They just considered that diabetes was a risk factor in PC development (Fisher, 2001). With the deeper understanding of the relationship between PC patient with diabetes and non-diabetes, recent data suggests that diabetes and altered in glucose metabolism are the consequence of PC, and yet, the clinical presentation of the altered glucose metabolism in these patients vary considerably (Yalniz & Pour, 2005). So, PC patients with diabetes and non-diabetes may represent two types of PC. Therefore, we predict that PC patients with diabetes and non-diabetes are also different in their prognostic biomarkers. The different prognostic biomarkers indicate that they should be treated respectively via their own different ways.

Generally, patients with diabetes have more opportunities to detect the potential risk of pancreatic cancer, while patients without diabetes often lack indicators for early diagnosis and miss the best opportunity for pancreatic cancer treatment. Furthermore, good prognostic markers can also be targeted at two types of pancreatic cancer patients to propose better treatment options, improve the prognosis.

In this study, The Cancer Genomic Atlas (TCGA) database, the Gene Expression Omnibus (GEO) database and the International Cancer Genome Consortium (ICGC)were employed to investigate and validate gene biomarker for prognosis in PC with or without diabetes. By characterizing genetic alterations, TCGA project has provided a large number of comprehensive genomic cancer data and corresponding clinical data that we can be used to figure out the relationship between them, which allows us to understand PC better and more accurate. However, high through-put genomic data (microarray or High seq V2) may encounter the problem in statistics which called “curse of dimensionality” (Mramor et al., 2005). Due to this problem, ordinary regression is subject to over-fitting and instable coefficients and stepwise variable selection methods do not scale well (Jr, Lee & Mark, 1996). Therefore, the least absolute shrinkage and selection operator (LASSO) method is employed to resolve this problem (Wang, You & Lian, 2015; Simon et al., 2011). Through adjusting the coefficient of Cox regression, LASSO can penalize the regression in high dimensionality and collinearity to solve “curse of dimensionality” (Tibshirani et al., 2012; Friedman, Hastie & Tibshirani, 2010). Least Absolute Shrinkage and Selection Operator (LASSO) regression and a hybrid of these (elastic net regression); all three methods are based on penalizing the L1 norm, the L2 norm, and both the L1 norm and L2 norm with tuning parameters. Although the traditional Cox proportional hazards model is widely used to discover cancer prognostic factors, it is not appropriate for the genomic setting due to the high dimensionality and collinearity. Several groups have proposed to combine the Cox regression model with the elastic net dimension reduction method to select survival-correlated genes within a high-dimensional expression dataset and have made available the associated computation procedures. Many studies have adopted elastic-net regression to screen genes, in order to predict survival of patients. In the current study, we are going to subject the integrated mRNA and clinical factors profiles of PC patients, aiming to identify and analyze gene biomarker that can predict the overall survival (OS) in the diabetes and non-diabetes of PC patients by LASSO.

Recently, many studies employed the TCGA (TCGA-PAAD) and GEO dataset (GSE62452) to identify useful gene biomarker which can predict prognosis in many various cancer patients (Bing et al., 2016; Yang et al., 2016). In this study, the ICGC dataset was also employed to validate prognostic gene signature. Along with the increasing genomic data of PC patients, lots of corresponding studies begin to analyze the genomic data and try their best to explore interesting and meaningful but extremely difficult problems (Gore et al., 2015; Craven et al., 2015).

Material and Methods

Information of patients

All related studies about diabetic and non-diabetic patients with PC were identified and collected by carefully searching from the online TCGA (TCGA: GDC TCGA Pancreatic Cancer) databases (http://tcga-data.nci.nih.gov/tcga/). The following combination of keywords was simultaneously applied for the literature search according to the requirement of this study ‘pancreatic cancer’ or ‘PC’ or ‘pancreatic tumor’ or ‘pancreatic malignancy’ and ‘diabetes’ and ‘non-diabetes’. In addition, the following research feature criteria are used to further improve and screen the desired search samples: (1) researches that concentrated on patients with diabetes and non-diabetes were selected; (2) survival time involved of patients was more than 30 days; (3) patients who didn’t receive any adjuvant therapy before. (4) all tissues that were from patients must be the primary tumor. After filtering and screening the data by these above criteria, 136 samples were selected from TCGA databases, which included 99 non-diabetic patients and 37 diabetic patients with PC.

RNA data gathering and filtering

The data of mRNA expression was downloaded from TCGA database. And the IIIumina HiSeq RNASeqV2 platform is selected.

Table 1 Clinical traits in PC patients with non-diabetes and diabetes.

	Non-diabetes PC(n = 99)	Diabetes PC(n = 37)	
Factors	Death/patients	Log-rank	Multivariate Cox P	Death/patients	Log-rank	Multivariate Cox P	
Age		0.051	0.496		0.959	0.446	
<=64	22/52			7/16			
>64	31/47			8/21			
Gender		0.402	0.172		0.001*	0.340	
Female	27/50			7/12			
Male	26/49			8/25			
Tumor Status		9.3e−06*	0.0004*		0.005*	0.513	
With Tumor	42/57			10/17			
Tumor Free	6/35			2/15			
Unknown	7/7			3/5			
Alcohol history		0.537	0.144		0.599	0.638	
Yes	40/68			10/27			
No	12/39			5/10			
Unknown	1/2			–			
History of chronic pancreatitis		0.597	0.998		0.273	0.998	
Yes	4/8			3/4			
No	48/86			10/31			
Unknown	1/5			2/2			
Number of lymph nodes positive by he		0.003*	0.396		0.480	0.533	
<3	22/52			7/20			
>=3	30/45			8/16			
Maximum tumor dimension		0.394	0.216		0.147	0.279	
>3.5	27/44			9/16			
<=3.5	26/51			6/20			
Neoplasm histologic grade		0.039*			0.004*		
G1	4/16		–	2/7		–	
G2	31/52		0.606	6/20		0.998	
G3	17/29		0.202	7/10		0.308	
G4	1/2		0.757	–		–	
TNM stage		0.100			0.431		
Stage I	0/1		–	0/1		–	
Stage IA	1/3		0.997	0/1		0.998	
Stage IB	3/10		0.998	0/2		0.998	
Stage IIA	5/13		0.998	3/7		0.998	
Stage IIB	43/70		0.998	11/24		0.998	
Stage III	1/2		–	0/1		–	
Stage IV	–		–	1/1		–	
Notes.

* p < 0.05, statistically significant.

Clinical factors and survival analysis

Clinical factors for the both diabetic and non-diabetic patients with PC are listed exhaustively in Table 1. For the correlation between RNA expression and OS was carried out by forthputting univariate Cox regression (the two-sided log-rank test). In the present meta-analysis, HRs and corresponding 95% CIs were combined to estimate the value of cancer prognosis. The hazard ratio (HR) was calculated from exp (β) and β was the coefficient from Cox regression. Clinical variables from univariate Cox proportional hazards regression P-value ≤0.05 were regarded as an important indicator of diabetic and non-diabetic patient prognosis.

The expression of mRNA associated with survival analysis

The relationship between patient survival and mRNA expression was analyzed through drawing on the univariate Cox proportional hazard regression. The null-selected RNA is calculated again and again. P-value ≤0.05 screened for mRNA (P ≤ 0.05). In normal conditions, RNAs that had a HR>1 and P value ≤0.05 were considered to be a risky gene while HR<1 is seen as an improved low-risky gene. In diabetic patients with PC, we reached a conclusion that 64 mRNAs are significantly associated with overall survival time (p < 0.05) by univariate Cox regression. In non-diabetic patients with PC, we acknowledged that 1,559 mRNAs are obvious significantly associated with overall survival time (p < 0.05). In data of high dimension gene expression, the coefficients (β) of Cox regression model needs to be penalized in order that it can fit better and minimize errors as much as possible. Therefore, elastic net-regulated Cox regression method is applied to calculate the results from univariate Cox regression. The penalized log-likelihood function is defined as following: lpβ,X=lβ,X−λ∑j=1p|βj|

With the value of λ increasing, value of ∑j=1p|βj| would be decreased. Then, some coefficients (β) of RNAs would be changed into 0. This result was analyzed by selecting the LASSO-adjusted Cox regression coefficient ≠0 mRNA. These steps are carried out by R package “glmnet”. Finally, we obtained eight mRNAs in diabetic patient with PC and 20 mRNAs in non-diabetic patients with PC.

Prognosis index construction

PI is calculated from linear combination of candidate RNAs and their expression for each PC patient. We defined a weighted prognostic index (WPI) (Xiong et al., 2014) for integrating indicators of RNAs for each PC patient, as following: (1) PI= ∑βi∗Vi

(2) WPI=PI−meanPISDPI

Where βi represents the coefficient in Cox regression of the i th variable. And Vi ii signifies the value of the i th variable. Mean (PI) and SD (PI) stand for the mean value and standard deviation of the PI, respectively. Where Vi is the expression value of each mRNA (log2-transformed expression value) and β i is the LASSO regulated Cox proportional hazards regression coefficient of the i th RNA or clinical traits.

Risk stratification and ROC curves

The capacity of the integrated RNA and clinical model to predict clinical outcome was evaluated by comparing the analysis of area under curve (AUC) of the receiver operation characteristic (ROC) curves. AUC for the ROC curve was applied to the “survival ROC” package in R software (Heagerty, Lumley & Pepe, 2000). The higher AUC is considered as a better model performance and range of AUC value is from 0.5 to 1. The AUC range from 0.80–0.90 is treated as good performance. And the range from 0.90–1.00 was considered to be excellent performance. The risk of patient group was classified into two groups based on the median of WPIs: high-risk and a low-risk. Survival analysis is forthputting Kaplan–Meier curves. Statistical analysis and graph in this study were performed using the software of R software (Ihaka & Gentleman, 1996), version 3.2.4 and Bioconductor, version 2.15 (Gentleman et al., 2004).

Gene ontology and pathway enrichment

Gene ontology (GO) functional enrichment analysis was performed to RNAs which classified as low-risk and high-risk group by making use of the online tool of the DAVID (version 6.8). We chose “Homo sapiens” as the background in order to search terms “GO_TERM_BP_FAT” for further analysis. And these genes are also enriched in Kyoto Encyclopedia of Genes and Genomes (KEGG) pathway for analysis (Aoki & Kanehisa, 2002).

Validation data of patient information collection

In this study, we selected two independent datasets to validation. An independent mRNA expression data of PC patients with 65 PC patients was downloaded from Gene Expression Omnibus (GEO: GSE62452) database (https://www.ncbi.nlm.nih.gov/geo/query/acc.cgi?acc=GSE62452). The clinical traits and expression were all downloaded from GSE62452. And the mRNA expression data were generated by Affymetrix Human Genome U133A Array. Data from GEO was analyzed using the updated July 26, 2018.

Another database was downloaded from ICGC database (https://dcc.icgc.org/). We selected Pancreatic Cancer –AU data for further validation. This dataset included 92 PC patients with RNAseq and clinical information. The genomic data of this dataset uses the technology of next generation sequencing. This gene data contained 56,026 RNAs and 92 patients’ follow-up data. We extracted gene signature from 56,026 RNAs for verification prognosis. (All raw data and code was listed in File S1).

Results

Clinical traits

In the TCGA PC cohort of the 136 patients, 99 patients are non-diabetic PC patients and 37 patients are diabetic PC patients. We calculated the clinical factors by adopting univariate survival analysis and multivariable Cox regression analysis. We selected nine clinical variables including age, gender, tumor status, alcohol history, history of chronic pancreatitis, number of lymph nodes positive, maximum tumor dimension, neoplasm histologic grade and pathologic stage. And these data are summarized in Table 1. In pancreatic patients without a diabetes cohort, tumor status was significantly associated with overall survival by long-rank and multivariate Cox regression analysis. This result indicated that tumor status is an independent factor correlated with overall survival. In pancreatic patients with diabetes cohort, gender is significantly associated with overall survival time. But this factor is not an independent factor by multivariate Cox regression analysis (Fig. 1, Table 1).

Figure 1 (A) Survival analysis in pancreatic cancer patient with non-diabetes. (B) WPI distribution in the TCGA pancreatic cancer cohort without diabetes. (C) Survival analysis in pancreatic cancer patient with diabetes. (D) WPI distribution in the TCGA pancreatic cancer cohort with diabetes.

Gene signature analysis in PC cohort

By analyzing of non-diabetes and diabetes PC patients through LASSO Cox regression and multivariate Cox regression, we have obtained 20 mRNAs and 8 mRNAs biomarkers, respectively, which were significantly associated with overall survival. Among these genes, the values of HR<1 and P value <0.01 were considered as protective RNAs and otherwise the values of HR >1 were risky RNAs (Tables 2 and 3). The graph for elastic net Cox regression can be found in Supplementary 1 and Supplementary 2.

Table 2 Gene biomarker in PC patients with non-diabetes.

	Hazard	95% CI	P-value	Description	
Low Risk genes	
TTTY9B	0	0.000–0.028	0.0102*	testis-specific transcript, Y-linked 9B (non-protein coding)	
RNF121	0.001	0.000–0.260	0.0142*	RING finger protein 121	
FHAD1	0.006	0.001–0.051	<0.001*	Forkhead-associated domain-containing protein 1	
GTF2F2	0.007	0.000–0.516	0.0235*	General transcription factor IIF subunit 2	
ADAMTS19	0.009	0.001–0.113	0.0002*	A disintegrin and metalloproteinase with thrombospondin motifs 19	
LHFPL1	0.024	0.002–0.283	0.0031*	Lipoma HMGIC fusion partner-like 1 protein	
DHDH	0.05	0.013–0.191	<0.001*	Trans-1,2-dihydrobenzene-1,2-diol dehydrogenase	
LOC256880	0.062	0.006–0.600	0.0164*		
SLC25A41	0.093	0.022–0.392	0.001*	Solute carrier family 25 member 41	
ZNF233	0.095	0.017–0.516	0.0060*	Zinc finger protein 233	
C6orf195	0.129	0.024–0.695	0.0171*		
PCDHA11	0.144	0.050–0.419	<0.001*	Proto cadherin alpha-11	
LOC401127	0.146	0.022–0.969	0.0463*		
TUBBP5	0.303	0.139–0.663	0.0028*	tubulin beta pseudo gene 5	
High risk genes	
CRCT1	2.107	1.154–3.847	0.0152*	Cysteine-rich C-terminal protein 1	
MUC20	14.76	4.387–49.66	<0.001*	Mucin-20	
RTP1	18.01	1.075–301.8	0.0444*	Receptor-transporting protein 1	
C10orf111	23.6	1.314–423.9	0.0319*		
SPACA5	23.83	1.821–311.7	0.0156*	Sperm acrosome-associated protein 5	
FZD10	26.54	5.142–136.9	<0.001*	Frizzled-10	
Notes.

* p < 0.05, statistically significant.

Table 3 Gene biomarker in PC patients with diabetes.

	Hazard	95% CI (95%)	p-value	Description	
Low Risk genes	
SYS1-DBNDD2	0.347	0.909–1.815	0.0020*		
NCRNA00167	0.231	0.978–1.719	0.0015*		
IRX5	0.473	0.282–1.185	0.0012*	Iroquois-class homeodomain protein IRX-5	
ZNF77	0.244	0.770–1.801	0.0040*	Zinc finger protein 77	
CATSPERG	0.296	0.651–0.991	0.0029*	Cation channel sperm-associated protein subunit gamma	
High Risk genes	
ZNF793	2.968	0.358–1.978	0.0063*	Zinc finger protein 793	
GBP6	1.744	0.342–1.207	0.0011*	Guanylate-binding protein 6	
FOSL1	2.306	0.9601–1.051	0.0091*	Fos-related antigen 1	
Notes.

* p < 0.05, statistically significant.

The PI was significantly associated with pancreatic patient survival. After normalized PI to WPI, the median value of WPI is acted as cutoff threshold to classify low-risk and high-risk patient cohort (Fig. 1).

Validation of the prognostic gene signature

The results were employed in two different ways to verify its stability and reliability. Firstly, we used the gene biomarker in PC patients with diabetes (8 mRNAs) to test the survival curve in PC patients with non-diabetes. Secondly, we used the gene biomarker in PC patients with non-diabetes (20 mRNAs) to swap above calculation.

The validated results showed that the gene biomarker in two groups performed poor result after exchange (Fig. 2). The results indicated that the gene biomarker in different groups has specificity in each condition.

Figure 2 Using gene signature of PC with diabetes to test in PC with non-diabetes.

(A) Gene signature of PC with diabetes validation in PC with non-diabetes. (B) Gene signature of PC with diabetes validation in PC with diabetes.

For validation result, independent mRNA expression data and corresponding clinical information of PC patient with non-diabetes is downloaded from GEO database to estimate the reproducibility and robustness of the results from TCGA database.

Gene ontology enrichment

The Database for Annotation, Visualization and Integrated Discovery (DAVID) v6.8 was employed to discover the function of genes both in PC patient with diabetes and non-diabetes. The eight genes in PC with diabetes were associated with regulation of transcription with a Benjamini–Hochberg correction P-value < 0.05. And many genes had DNA binding function. For 20 genes identified in PC without diabetes were not enriched statistically significant association.

Comparison of clinical traits and gene biomarker for predicting prognosis

We integrated clinical traits that significantly associated with survival and PI of gene biomarker that significantly associated with survival to analyze the pancreatic cancer in diabetic and non-diabetic individuals. After multivariate Cox regression analysis, the results showed that PI of gene biomarker performed greatest P-value (Table 4). We filtered the clinical factors that significantly associated with survival by log-rank test into integrative model. In PC with non-diabetes, tumor status, number of lymph nodes positive, stage G2, G3 and G4 were significantly associated with survival (Table 1). And in PC with diabetes, gender, stage G2 and G3 were significantly associated with survival by log-rank test (Table 1).

Table 4 Multivariate Cox regression analysis of prognosis index and clinical traits.

PC with Non-diabetes	HR	CI	Multivariate Cox P-value	
PI	1.102	1.070–1.136	2.68e−10*	
Tumor Status	0.117	0.298–1.924	0.0005*	
Number of lymph nodes positive by he	1.589	0.907–2.783	0.106	
G2	2.103	0.187–5.400	0.123	
G3	2.036	0.739–5.613	0.169	
G4	2.215	0.257–19.087	0.469	
PC with Diabetes				
PI	1.212	1.108–1.327	2.83e−05*	
Gender	0.173	0.053–0.564	0.004*	
G2	0.897	0.168–4.775	0.898	
G3	5.310	0.892–31.616	0.067	
Notes.

* p < 0.05, statistically significant.

From the Table 4, we find PI of gene biomarker have smallest P-value after multivariable Cox regression. Although HR is not the highest among clinical traits, P-value is the smallest. Besides, we can find that tumor status is another significant risk factor in PC with non-diabetes.

Independent data validation for PC with non-diabetes

For further validation result, independent mRNA expression data and corresponding clinical information of PC patient with non-diabetes is downloaded from GEO database (GSE62452) to estimate the reproducibility and robustness of the results from TCGA database. The results showed that the gene signature from TCGA data could be validated in GEO database (n = 65). PI was calculated from gene signature can effectively predict survival of PC with non-diabetes. The median of PI value divided 65 patients into high-risk group and low-risk group (HR = 3.006, P-value < 0.001). And results of ROC showed that AUC = 0.828. The results indicated that the gene signature from TCGA could be validated in independent dataset (Fig. 3).

Figure 3 The gene biomarker can greatly classifiy PC patients into high-risk and low-risk groups (p < 0.001).

The AUC of ROC is 0.828, which represent that the gene biomarker model is very good. (A) Risk and overall survival in GEO validation cohort; (B) ROC in GEO validation cohort.

Pancreatic cancer data was downloaded from ICGC database. This data included 92 patients with genomic data and clinical information. The gene signature was matched ICGC database and constructed PI model. The results showed that the PI from gene signature can divided patients into high-risk and low-risk groups significantly (HR = 2.84, P-value < 0.001) in ICGC data. ROC showed that AUC = 0.74, which indicated that the gene signature also validated in ICGC and predict performance well in 3 years (Fig. 4).

Figure 4 The gene signature validated in the ICGC database.

(A) Risk and overall survival in the ICGC validation cohort; (B) ROC in the ICGC database validation.

Discussion

PC patients showed different prognostic gene signature in diabetes and non-diabetes. Identification special gene signature in different types of PC patients would provide precise medicine for different patients. We identified and verified specific high-risk genes for PC patients without diabetes. And these genes have not been reported before. These gene targets may be potential therapeutic targets for pancreatic cancer.

In this study, we proposed two classes of gene biomarkers in PC patients with and without diabetes which can guide us to predict PC patient survival better and more accurate. To a large extent, PC patients with and without diabetes have quite different gene biomarker for predicting prognosis. After a series of studies, we not only find that genes candidate in both PC patient groups have no overlapping but also figure out that gene biomarker in non-diabetes PC patients is validated by GEO and ICGC datasets. The result indicated that the two sets of gene biomarker in both groups have been very specified. Therefore, they have their own gene biomarker for predicting their prognosis. Because the differences between diabetic and non-diabetic pancreatic cancer patients are often ignored, we only got two types of patients in TCGA database. Other validation databases contained only non-diabetic patients. Furthermore, non-diabetic patients with pancreatic cancer are more likely to be ignored in the diagnosis, leading to a higher risk of such patients. Thus, we validated gene biomarker in non-diabetes PC patients in more datasets. Although a large number of studies have reported some biomarkers in PC patients, many genes have been identified primarily in PC patients without diabetes. We identified and compared the gene signature that predict both types of PC patients. And many genes have not been reported yet so far. Among the high risk prognostic genes, CRCT1, MUC20, RTP1, C10orf111, SPACA5 and FZD10 have high level of HR. MUC20, FZD10 have been identified in PC patients (Lee et al., 2016; Kirikoshi & Katoh, 2002) and these two genes play a vital role in two important pathways associated with cancer. MUC20 is involved in MET (Mesenchymal-Epithelial transitions) process which is a common process in many tumors (Spaderna et al., 2007). And it may regulate MET signaling cascade. It appears to decrease hepatocyte growth factor (HGF)-induced transient MAPK activation (Higuchi et al., 2004). FZD10 is associated with WNT signaling pathway which is implicated in embryogenesis as well as in carcinogenesis (Terasaki et al., 2002). Other genes were not reported in PC patients, but only SPACA5 is reported in bladder cancer (Zhang, Chen & Chen, 2016). Although many genes have not been reported before, we find that these combinations of these genes can greatly distinguish high-risk and low-risk PC patients with non-diabetes. In addition, these genes were validated in an independent GEO database and ICGC database. The results of GSE62452 in the GEO database indicated that these genes were stably expressed and the gene biomarker could distinct between high-risk and low-risk gene greatly.

The gene biomarker in PC patients with diabetes, three genes are high-risk genes. We can find that the production of these three genes (ZNF793, GBP6, FOSL1) are binding function proteins. Thus, we infer that they are all transcription factors. Of the three genes, FOSL1 has been reported to be closely associated with PC (Vallejo, Valencia & Vicent, 2017; Vallejo et al., 2017; Sahin et al., 2005). But these studies have not reported that this high-risk gene is associated with PC with diabetes yet. Only one study reported that FOSL1 is closely associated with diabetes mellitus (Portal-Núñez et al., 2010). And this gene has not been identified in PC with non-diabetes. GBP6 is reported in diabetes (O’Tierney et al., 2012) but is not reported in PC patients with diabetes. ZNF793 is not identified in both PC and diabetes. Thus, we infer that the gene is a potential risk factor in PC patients with diabetes.

Through multivariate Cox regression analysis, it is interesting to note that tumor status is an independent predictor of prognosis in non-diabetes PC patients. Gender is an independent predictor of prognosis in patients with diabetes in PC. Tumor status is a vital clinical factor for predicting the prognosis in many cancers.

From the results, we find that there was no overlapping of both groups. Thus, we conclude that two types of PC vary greatly at the molecular level. Prognostic gene signature in non-diabetes PC patients showed robustness among two datasets (GEO and ICGC). Many genes have not reported in publication and we hope that these genes can predict prognosis for improving clinical decision.

Conclusion

Pancreatic cancer patients with diabetes and without diabetes have different gene signature for predicting their respective prognosis. The results indicated that the gene signature of pancreatic cancer patients without diabetes has been validated in two independent datasets. Thus, the different gene marker might be as an useful tool for clinical decisions in the future.

Supplemental Information

Supplemental Information 1 Raw data and code for gene expression analysis

Click here for additional data file.

Supplemental Information 2 The cross-validation error curve of pancreatic cancer with diabetes

Click here for additional data file.

Supplemental Information 3 The cross-validation error curve of pancreatic cancer with non-diabetes

Click here for additional data file.

Additional Information and Declarations

Competing Interests

Author Contributions

Data Availability

The authors declare there are no competing interests.

Mingjun Yang conceived and designed the experiments, authored or reviewed drafts of the paper, and approved the final draft.

Boni Song analyzed the data, prepared figures and/or tables, authored or reviewed drafts of the paper, and approved the final draft.

Juxiang Liu and Yonggang Wang performed the experiments, prepared figures and/or tables, and approved the final draft.

Zhitong Bing conceived and designed the experiments, analyzed the data, prepared figures and/or tables, authored or reviewed drafts of the paper, and approved the final draft.

Linmiao Yu performed the experiments, authored or reviewed drafts of the paper, and approved the final draft.

The following information was supplied regarding data availability:

The raw measurements are available in the Supplemental Files.

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
