# Peer review of "Gene signature for prognosis in comparison of pancreatic cancer patients with diabetes and non-diabetes"

_PeerJ, doi:10.7717/peerj.10297_

## Round 0.1 · original submission · Major Revisions

The manuscript needs extensive English editing because there are several typos and grammatical errors. The reviewers indicated that the language used in the manuscript should be improved. I agree with this evaluation and I would, therefore, request for the manuscript to be revised accordingly especially interpreting statistical analysis results. In addition to this:

In Line 83 and Line 153, the access date of database and accession numbers of datasets must be given.

In Line 134, provide more details according to the LASSO regulated Cox proportional hazards regression.

In Line 177, it is not clear why the authors prefer to use the median value as cut-off?

In Line 193, Is that “Benjamini correction” refer to “Benjamini-Hochberg correction” or “Bonferroni correction”? Please revise this.

The authors did not provide detailed results of ROC analysis and not discussed these results with p-values and AUC values. Provide details of their findings, and add these figures in the manuscript.

For all tables with p-values, please use * for p<0.05 and write the phrase “*p<0.05, statistically significant” in the footnote of the table. Tables are not readable. Improve readability.

Reviewer 1 ·

Basic reporting

I would recommend that the authors seek professional language services to improve their writing.

The terms ‘protective’ and ‘risky’ genes may be replaced with more suitable terms. Please refer to other articles describing prognostic biomarkers.

The use of ‘generally speaking’ should be avoided.

Experimental design

Experimental design appears to be appropriate.

Validity of the findings

The authors identified a set of prognostic biomarkers for pancreatic cancer patients from TCGA. They have validated these markers using a dataset from GEO. It would be useful to validate the markers in another dataset (with more patient numbers) from ICGC.

Reviewer 2 ·

Basic reporting

The English language should be improved to ensure that an
International audience can clearly understand your text.

The authors do not discuss the clinical significance of their findings and the utility and nature of markers they found. The markers are correlative and not necessarily causative.

The conclusion section needs to be improved and the authors need to elaborate what are the main findings of their study.

Specific comments are added in the annotated PDF file.

Experimental design

no comment

Validity of the findings

Impact and novelty not discussed
Conclusions are not well stated and need to be rewritten

Annotated reviews are not available for download in order to protect the identity of reviewers who chose to remain anonymous.

Reviewer 3 ·

Basic reporting

The manuscript is unfortunately poorly written. The use of Scientific English and the grammar is poor. Throughout the manuscript the given information, findings and discussion of the data is not intelligible.

Writers do not link their hypothesis and their findings in the text and often make confusing jumps to other findings. The coherence is severely lacking.

Experimental design

Clinical findings on tumor state and histological grading is not novel, what authors here try to show is the difference in diabetes groups however there is no highlight to diabetes and in the manuscript it feels like they are repetitively mentioning well established findings.

The results and their discussion should be focused on the novel findings.

Validity of the findings

There are mislabeled figures and table numbers in the main text body, which is misleading and confusing (e.g in lines 203 and in 204 the label says Table 2 however it must be Table 1)

Some findings that are essential to the hypothesis are completely untouched, not mentioned.

Authors should give more conscience explanation of each result.

Additional comments

I advise for thorough editing of the use of English. From the introduction to the discussion it is very hard to understand and follow the logic of the findings.

also more detailed explanation of the rationale of the study and the findings must be given.

Reviewer 4 ·

Basic reporting

pass

Experimental design

pass

Validity of the findings

pass

Additional comments

The authors reported that the integrated gene prognostic biomarker systems are identified in PC with non-diabetes or diabetes

1. Are there relation or differences between two prognostic biomarker systems?
2. The diabetes were newly discovered in PC patients?was there any difference between the newly and old diabetes?
3. TNM staging is most important for prognosis, why the author did not include TNM staging?

---

## Round 0.2 · accepted · Accept

The authors addressed the reviewers' concerns and substantially improved the content of MS. So, based on my own assessment as an academic editor, no further revisions are required and the MS can be accepted in its current form.

Reviewer 2 ·

Basic reporting

no comment

Experimental design

no comment

Validity of the findings

no comment

Additional comments

no comment

Reviewer 3 ·

Basic reporting

This manuscript presents an interesting analysis of publicly available PDAC datasets and presents a gene signature that can be utilized for a prognostic assessment. However, the use of academic English is needed to be enhanced for better communication. An overall grammar check might also give the manuscript better readability.

Experimental design

1.Throughout the manuscript, authors refer to different gene signatures as "gene biomarker", where there are two different sets used. A more specific naming for these gene signatures, for example, Diabetes-PC or non-Diabetes-PC, must be given to maintain the readers' attention and understanding.

2. A heatmap for differential expression between Diabetes and Non-diabetes must be given for proposed gene biomarkers, in addition to LASSO cox regression. The expression levels of proposed genes are equally important for the overall analysis. This must be further added to the discussion.

Validity of the findings

no further comment